# Design Optimization of a Multi-Megawatt Wind Turbine Blade with the NPU-MWA Airfoil Family

**Jianhua Xu**, **Zhonghua Han \***, **Xiaochao Yan and Wenping Song**

Institute of Aerodynamic and Multidisciplinary Design Optimization, National Key Laboratory of Science and Technology on Aerodynamic Design and Research, School of Aeronautics, Northwestern Polytechnical University, Xi'an 710072, China

\* Correspondence: hanzh@nwpu.edu.cn

**Abstract:** A new airfoil family, called NPU-MWA (Northwestern Polytechnical University Multi-megawatt Wind-turbine A-series) airfoils, was designed to improve both aerodynamic and structural performance, with the outboard airfoils being designed at high design lift coefficient and high Reynolds number, and the inboard airfoils being designed as flat-back airfoils. This article aims to design a multi-megawatt wind turbine blade in order to demonstrate the advantages of the NPU-MWA airfoils in improving wind energy capturing and structural weight reduction. The distributions of chord length and twist angle for a 5 MW wind turbine blade are optimized by a Kriging surrogate model-based optimizer, with aerodynamic performance being evaluated by blade element-momentum theory. The Reynolds-averaged Navier–Stokes equations solver was used to validate the improvement in aerodynamic performance. Results show that compared with an existing NREL (National Renewable Energy Laboratory) 5 MW blade, the maximum power coefficient of the optimized NPU 5 MW blade is larger, and the chord lengths at all span-wise sections are dramatically smaller, resulting in a significant structural weight reduction (9%). It is shown that the NPU-MWA airfoils feature excellent aerodynamic and structural performance for the design of multi-megawatt wind turbine blades.

**Keywords:** aerodynamic design optimization; wind turbine; airfoil; flat-back airfoil

---

## 1. Introduction

Among the types of renewable energies, wind energy has the advantages of large reserves, cleanliness, low exploitation cost, and high reliability. Wind energy is also the best renewable energy that is able to be exploited on a large scale and utilized on the market at present, and is favored by people from all over the world. Many countries have formulated wind energy exploitation plans and established state-funded research centers and laboratories to carry out related research. According to the statistics from Global Wind Energy Council (GWEC) [1–4], the yearly newly installed capacity of wind power in the world has exceeded 50 GW since 2014, with a yearly increase of 10% in the new installed capacity, and it is predicted to reach an installed capacity of 840 GW in 2022.

The aerodynamic performance of airfoils has a significant influence on the aerodynamic characteristics of a wind turbine blade. Before 1990, wind turbine blades were designed by using the conventional aeronautic airfoils, such as the NACA (National Advisory Committee for Aeronautics) 6-series (NACA63, NACA64, etc.) and four-digit airfoils (NACA 44-series). Since the late 1980s, research on advanced airfoils dedicated to wind turbines has been carried out in the United States of America and Europe, such as the S-series airfoils designed by National Renewable Energy Laboratory (NREL) of the United States [5,6], the DU-series airfoils designed by TU Delft of the Netherlands [7], the RISØ-series airfoils designed by RISØ Laboratory of Denmark [8,9], and the FFA-series airfoils

designed by National Aeronautical Research Institute of Sweden [10]. The Northwestern Polytechnical University (NPU) of China has developed a family of NPU-WA (Northwestern Polytechnical University Wind-turbine A-series) airfoils for megawatt wind turbines, featuring a high lift-to-drag ratio at a high Reynolds number and a high lift coefficient [11–14]. The Chinese Academy of Sciences (CAS) has designed a family of CAS airfoils for medium Reynolds numbers [15]. The Chongqing University of China and Technical University of Denmark have developed the CQU-DTU-series airfoils, taking into account of the aerodynamic performance and low noise requirements [16]. Compared with the conventional aeronautic airfoils, the dedicated airfoils for wind turbine blades have a higher lift-to-drag ratio, which means that the aerodynamic force of the blades has a larger component in the tangential direction of the wind wheel and lower component in the axial direction (thrust), resulting in smaller thrust with the same tangential component of aerodynamic force, thus the total aerodynamic load is reduced.

With the continuous development of wind turbine technology, the wind turbine generator system tends to become larger and larger. The blade diameter of wind turbines has increased from about 20 m in the 1980s to nearly 200 m, or even more now. Meanwhile, the capacity has also been increased, from 20 kW~60 kW to 5 MW~10 MW [17–19]. In recent years, the technology for applying the wind turbine-dedicated airfoil family to megawatt wind turbines (1.0 MW, 3.0 MW) has gradually matured. More and more attention has been paid to the development of new airfoil families dedicated to larger wind turbines, called multi-megawatt wind turbines (3 MW, 10 MW), especially for offshore wind energy applications. Some companies have formed a production capacity of 3 MW–5 MW wind turbines, such as EP 5 Series [20] of Enercon and the 4 MW platform [21] of Vestas. In addition, the design processes of the 10 MW wind turbines in Vestas, the 7 MW wind turbines in Enercon, and the 6 MW wind turbines in the DOWEC (Dutch Offshore Wind Energy Converter) are ongoing.

The rapid development of multi-megawatt wind turbines has inspired the research and design of new airfoil families. According to the operating conditions of multi-megawatt wind turbine blades, new challenges have arisen for the design and verification of new airfoil families. First, the airfoils dedicated to multi-megawatt wind turbine blades are expected to have excellent aerodynamic performance at even higher design lift coefficient and higher Reynolds number, and have higher maximum lift coefficient than the airfoils dedicated for megawatt wind turbine blades. The operating Reynolds number of the main airfoil can easily reach at least 9 million, and the design lift coefficient can be larger than 1.2. Second, structural safety is particularly prominent due to increased blade size and structural weight [22]. For inboard airfoils, special tailored flat-back airfoils [23,24] with very large trailing-edge thickness should be developed to promote the blade structural performance and at the same time to take into account the high-lift aerodynamic characteristics at a price of increased drag.

Supported by the National High Technology Research and Development Program ("863" Program) of China, a new airfoil family, named NPU-MWA airfoils, for multi-megawatt wind turbine blades was designed by Han et al. [13,14], and the invention patent of China [25] has been authorized. The performance of these airfoils has been validated by the NF-3 low-speed wind tunnel of Northwestern Polytechnical University. The test results show that the designed airfoils have excellent aerodynamic performance, with high design lift coefficients (>1.2) and high lift-to-drag ratio at high Reynolds number. The inboard flat-back airfoils enable the blade to have better structural performance than conventional sharp-trailing edge airfoils.

The objective of this article is to demonstrate the advantages of the newly developed NPU-MWA airfoils in improving wind energy capturing and structural weight reduction, by carrying out an aerodynamic optimization design of 5 MW wind turbine blades. The remainder of this article is organized as follows. In Section 2, the developed NPU-MWA airfoil family is introduced, including design goals and philosophy, geometric shapes and characteristics, and the aerodynamic performance test in the NF-3 wind tunnel. Section 3 describes the design optimization of multi-megawatt wind turbine blades, including an aerodynamic analysis model and validation, mathematical model of optimization, and efficient surrogate-based optimizer. Section 4 presents the results and discussion,

including the design optimization of multi-megawatt wind turbine blades with NPU-MWA airfoils, and a comparison of aerodynamic and structural performance between the optimized NPU 5 MW blade and the existing NREL 5 MW blade. The last section is for the conclusions.

## 2. NPU-MWA Airfoil Family

In 2016, a new airfoil family, NPU-MWA airfoils, was designed for multi-megawatt wind turbines by Han et al. [14], and the invention patent of China [25] was authorized. In this section, the airfoil family is briefly introduced and then we will use the airfoil family to design a multi-megawatt wind turbine blade. Figure 1 shows the design goals of NPU-MWA airfoils at different radius sections [26]. The importance at different spanwise of the blade for aerodynamic performance and structural performance is reversed. The most important goal for airfoils outboard of the blade is high aerodynamic performance, such as high maximum lift-to-drag ratio, benign post stall, and also low noise. However, the most important goal for airfoils inboard of the blade is high structural performance. In addition, geometric compatibility is equally important for all airfoils.

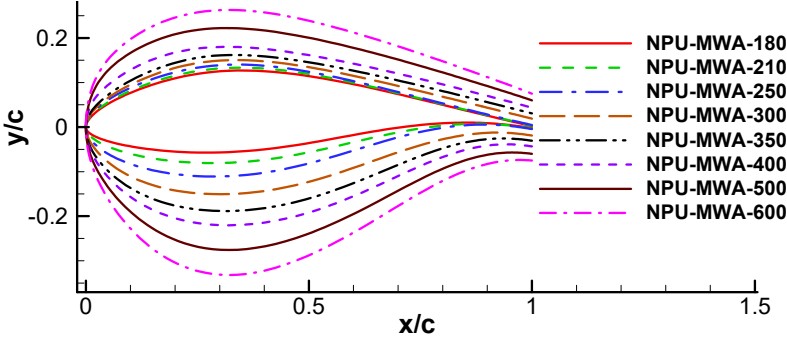

|  | Inboard | Mid span | Outboard |
|---|---|---|---|
| **High maximum lift-to-drag ratio** | √ | √√ | √√√ |
| **Insensitivity to roughness** | √ | √√ | √√√ |
| **Geometric compatibility** | √√ | √√ | √√ |
| **Structural demands** | √√√ | √√ | √ |
| **benign post stall** |  |  | √√ |
| **Low noise** |  | √ | √√ |

**Figure 1.** Design goals of NPU-MWA airfoils (created referring to [26], more √ indicates greater importance).

The geometric shapes of the airfoils are shown in Figure 2. These airfoils are designed for achieving a high lift-to-drag ratio at high design lift coefficients (>1.2). A high lift-to-drag ratio helps to improve the aerodynamic performance and wind energy capturing of a wind turbine. A high design lift coefficient is beneficial for reducing the chord length of a blade, resulting in a structural weight reduction. The flat-back airfoils adopted for inboard of the blade are also helpful for improving the structural performance by reducing structural weight. The design Reynolds numbers for airfoils at outboard of the blade reach up to 9 million, such as NPU-MWA-250, NPU-MWA-210, and NPU-MWA-180. The design Reynolds numbers for airfoils inboard of the blade decreases gradually, to a minimum value of 4 million.

**Figure 2.** Geometric shape of the NPU-MWA wind turbine airfoil family.

The aerodynamic performance of NPU-MWA wind turbine airfoils was validated by the NF-3 low-speed wind tunnel of Northwestern Polytechnical University (Figure 3). Shown in Figures 4 and 5, compared to DU airfoils, the NPU-MWA airfoils have a higher maximum lift coefficient, higher lift-to-drag ratio, and higher design lift coefficients (corresponding to the maximum lift-to-drag ratio). It is demonstrated that the design lift coefficient is higher than 1.2. It is noted that the stall of the 21% thick airfoil is much more abrupt than the DU airfoil. Shown in Figure 6, the sensitivity of the maximum lift coefficient to leading edge roughness is smaller than 15% over a wide range of Reynolds numbers, indicating that the maximum lift coefficient of the NPU-MWA airfoils has low sensitivity, leading to edge roughness. The blade is easily contaminated by dust, insects, sand, and seawater, which changes the shape of the blade and may cause serious deterioration of aerodynamic performance [27]. The low roughness sensitivity of airfoils means the aerodynamic performance of the contaminated blade does not decrease obviously, compared with a smooth blade.

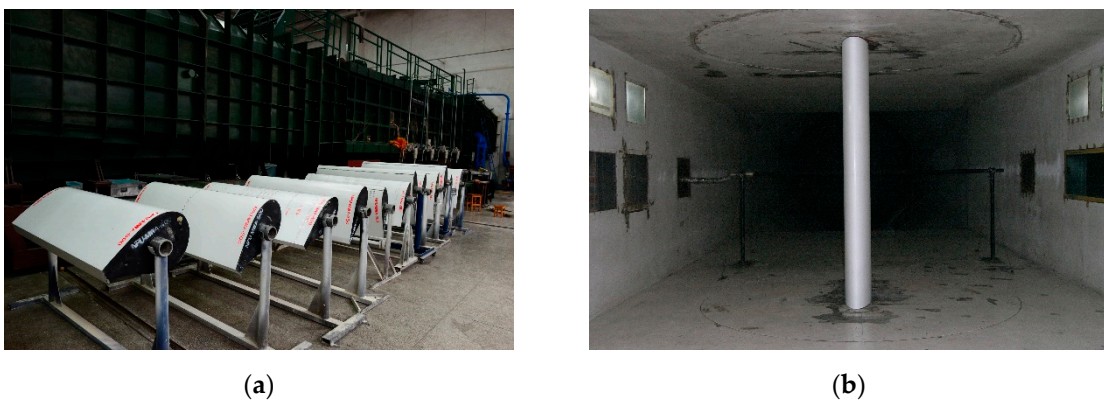

(a)　　　　　　　　　　　(b)

**Figure 3.** Experimental test of NPU-MWA wind turbine airfoils in the NF-3 low-speed wind tunnel of Northwestern Polytechnical University: (**a**) test models of NPU-MWA airfoils; (**b**) model in the test section of NF-3 wind tunnel.

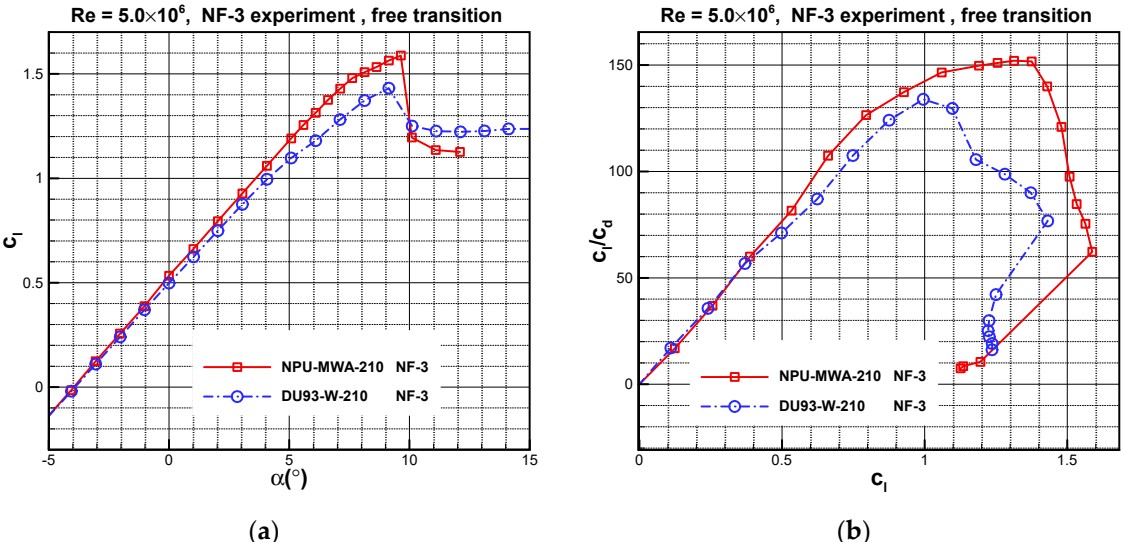

(a)　　　　　　　　　　　(b)

**Figure 4.** Comparison of experimental data for the NPU-MWA-210 airfoil and DU93-W-210 airfoil: (**a**) lift curves; (**b**) lift-to-drag ratio curves.

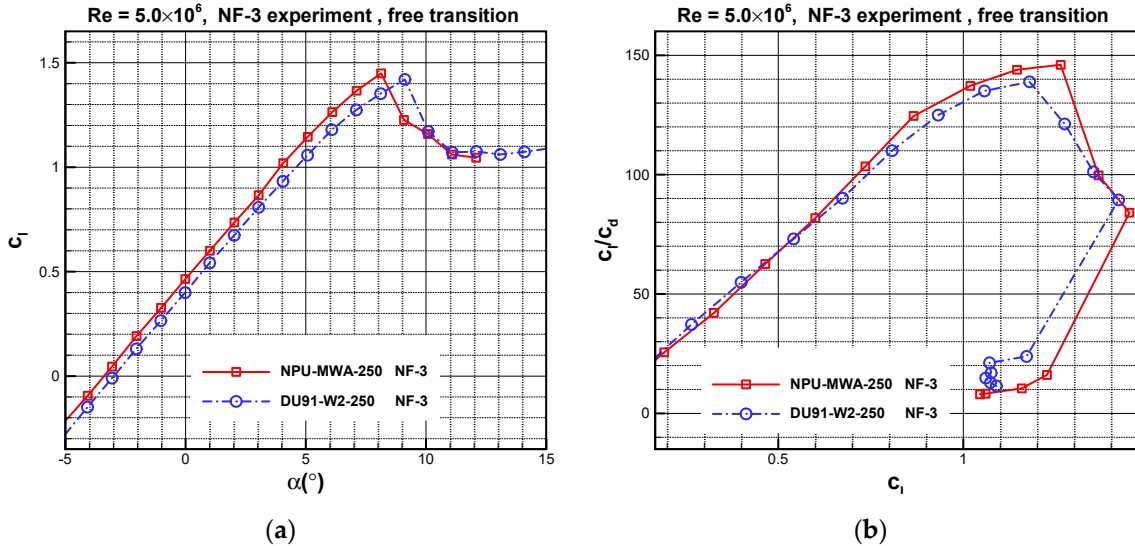

**Figure 5.** Comparison of experimental data for the NPU-MWA-250 airfoil and DU91-W2-250 airfoil: (**a**) lift curves; (**b**) lift-to-drag ratio curves.

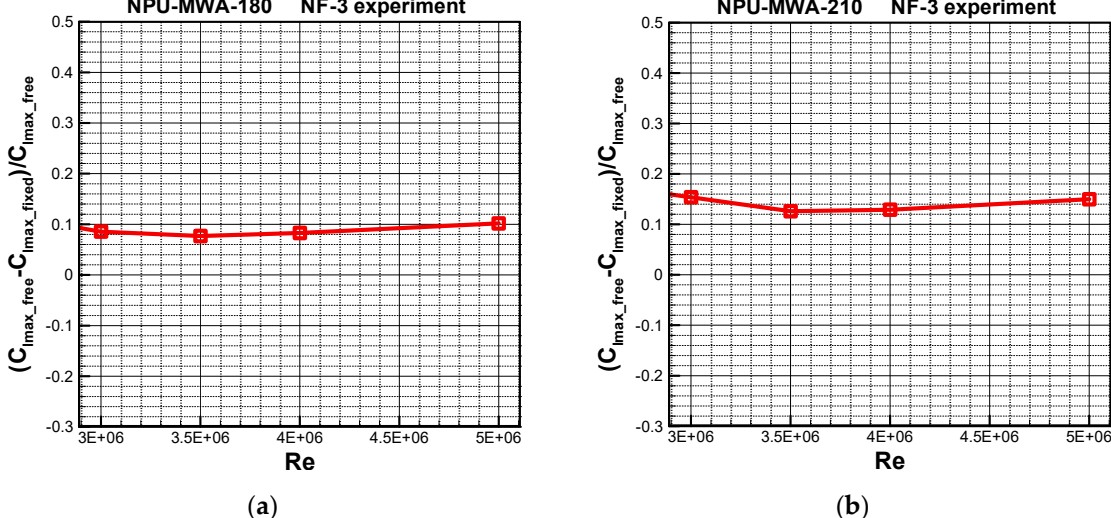

**Figure 6.** Sensitivity of the maximum lift coefficient to the leading edge roughness with the increase of Reynolds number (Re): (**a**) NPU-MWA-180 airfoil; (**b**) NPU-MWA-210 airfoil.

In summary, the NPU-MWA airfoils have excellent aerodynamic performance, structural characteristics, and geometric compatibility, which is helpful in improving the aerodynamic and structural characteristics of the blade, while reducing the structural weight of the blade.

## 3. Design Optimization of a Wind Turbine Blade

Computational fluid dynamic (CFD) methods, including the Reynolds-averaged Navier–Stokes (RANS) simulation [28], delayed detached eddy simulation (DDES) [29], and large eddy simulation (LES) are high fidelity tools for wind turbine blade design. However, they are too consuming for an optimization procedure. Instead, the blade element-momentum (BEM) theory [30,31], which is of much lower cost and gives reasonable results, is widely adopted in realistic engineering applications [32,33]. In the present study, the BEM theory is used to evaluate the aerodynamic performance for the optimization design of a wind turbine blade, and an in-house RANS solver is used to validate the optimal results, and provide reliable aerodynamic loads for structural analysis.

### 3.1. Aerodynamic Analysis Models and Validation

The BEM theory is the most commonly used method in the aerodynamic analysis of a wind turbine blade. Some well-known software for wind turbine blades design and analysis are based on this method, such as GH Bladed [34] and PROPID [35].

The RANS method can better simulate the viscous effect, circulation variation, and evolution of wake vortices of flow over a three-dimensional wind turbine blade, and gives more description of flow structure and details [36,37]. In the present work, an in-house code "ROTNS" [38] was used to simulate the quasi-steady flow around wind turbine blades. In this code, the cell-centered finite-volume method developed by Jameson [39] was used to solve the RANS equations on the rotational coordinate system fixed to the blade. Jameson's central scheme [39] was utilized for the spatial discretization scheme, the lower-upper symmetric-Gauss–Seidel (LU-SGS) [40] scheme was for implicit time stepping, and the Spalart–Allmaras (S-A) [41] one-equation model was used for turbulence enclosure.

Only one blade was considered in the simulation and other blades were described by the periodic boundary conditions. The chimera grid system based on two grids in the simulation is involved and sketched in Figure 7, and included a background grid and a blade grid, both of which moved with the rotary blade. The background grid (H-H type) was for far-field simulation, and the periodic boundary condition was implemented in a point to point manner.

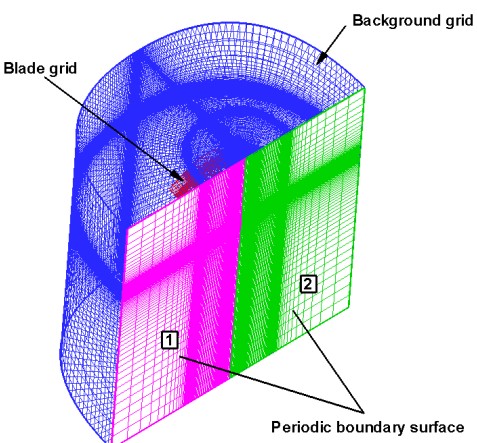

**Figure 7.** Sketch of the chimera grid system and periodic boundary conditions.

The BEM method and RANS solver used in this study were validated by the case of the Phase VI wind turbine blade designed by American National Renewable Energy Laboratory (NREL) [28,42,43]. The wind turbine consisted of two blades, and the blade radius was 5.03 m. Under the stall-control mode, the rated power was 19.8 kW, corresponding a rotational speed of 72 rpm. The detailed geometric information refers to Reference [42]. The chord length and twist angle distributions are shown in Figure 8a and the configuration is presented in Figure 8b.

Simulations were conducted at a rotational speed of 72 rpm and wind speeds of 7 m/s, 10 m/s, 13 m/s, 15 m/s, 20 m/s, and 25 m/s. The experimental conditions and results refer to Reference [1]. Four sets of grids were used in the RANS simulation, shown in Table 1.

**Table 1.** Four sets of grids used in the Reynolds-averaged Navier–Stokes (RANS) simulation.

|  | **Blade Grid** | **Background Grid** | **Total Grid Points** |
|---|---|---|---|
| Grid 1 | $185 \times 81 \times 85$ (1,273,725) | $101 \times 253 \times 121$ (3,091,913) | 4,365,638 |
| Grid 2 | $225 \times 101 \times 85$ (1,931,625) | $101 \times 301 \times 121$ (3,678,521) | 5,610,146 |
| Grid 3 | $225 \times 129 \times 125$ (3,628,125) | $101 \times 331 \times 121$ (4,045,151) | 7,673,276 |
| Grid 4 | $225 \times 161 \times 165$ (5,977,125) | $101 \times 331 \times 121$ (4,045,151) | 10,022,276 |

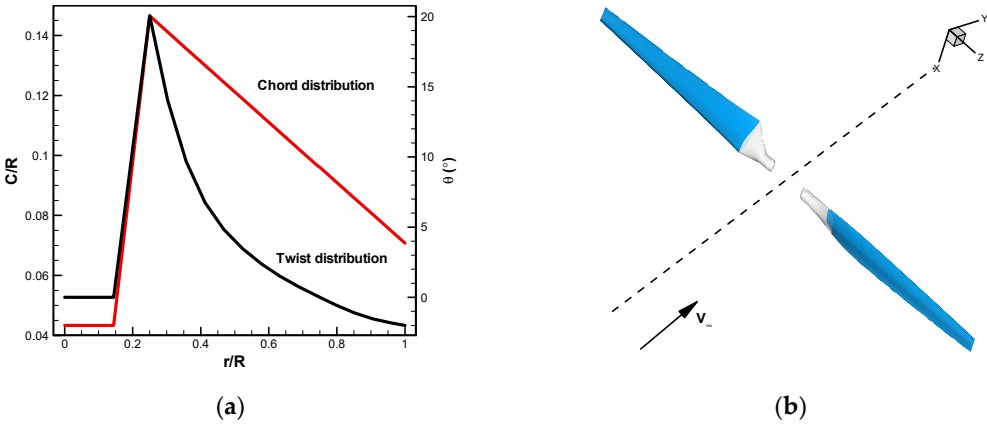

(**a**)                                           (**b**)

**Figure 8.** National Renewable Energy Laboratory (NREL) Phase VI wind turbine blade: (**a**) distributions of chord length and twist angle; (**b**) geometric shape of blade.

Figure 9 shows that the BEM method gave fine results. Also, the present results tended to converge with an increase in grid size and agree well with the experimental data, which were better than the RANS results referred to in Reference [1]. As the results of Grid 3 and Grid 4 were almost consistent, the following simulations keep the same size as Grid 3. Figures 10–12 show good agreement of pressure distributions between experimental data and present results at the wind speeds of 7 m/s, 10 m/s, and 20 m/s, respectively.

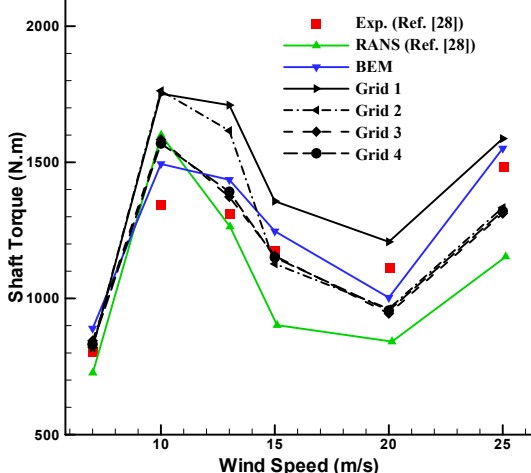

**Figure 9.** Comparison of experimental data, reference results, and present results of shaft torque.

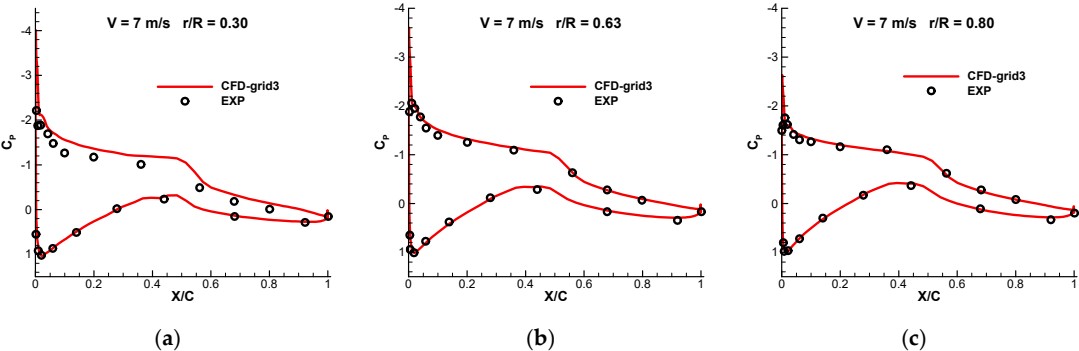

(**a**)                                           (**b**)                                           (**c**)

**Figure 10.** Comparison of pressure distributions between experimental data and present results (wind speed 7 m/s): (**a**) r/R = 0.30; (**b**) r/R = 0.63; (**c**) r/R = 0.80.

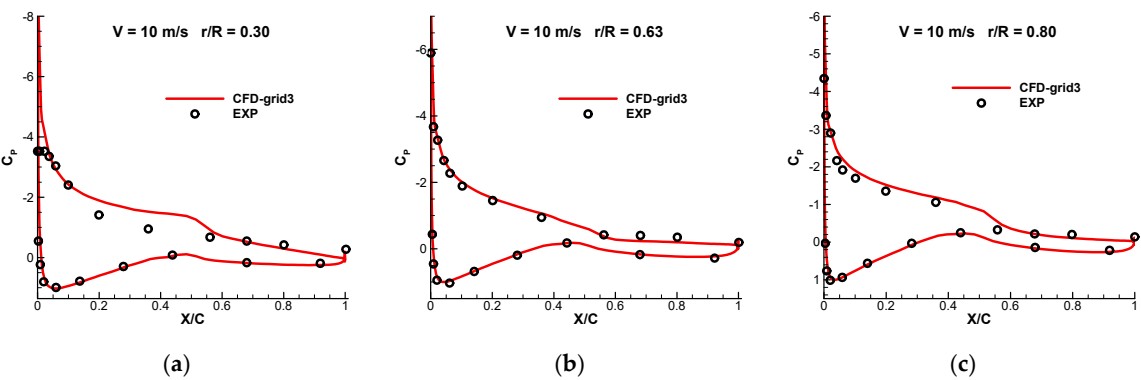

**Figure 11.** Comparison of pressure distributions between experimental data and present results (wind speed 10 m/s): (**a**) r/R = 0.30; (**b**) r/R = 0.63; (**c**) r/R = 0.80.

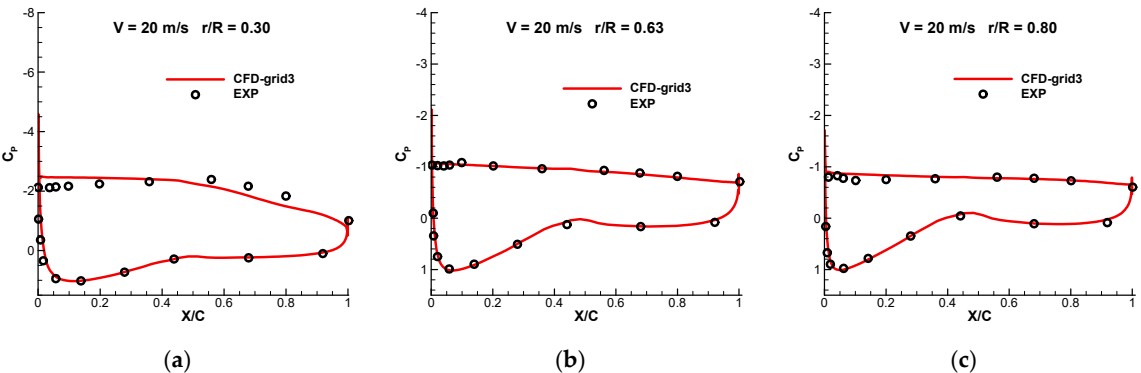

**Figure 12.** Comparison of pressure distributions between experimental data and present results (wind speed 20 m/s): (**a**) r/R = 0.30; (**b**) r/R = 0.63; (**c**) r/R = 0.80.

### 3.2. Design Optimizatoin Method

The design goal was to maximize the maximum power coefficient ($CP_{\max}$), the design variables were chord lengths and twist angles at 17 control sections, and the constraints included the optimal tip-speed ratio, power coefficients at a high tip-speed ratio, chord, and twist of control sections. The mathematical model of optimization can be expressed as follows:

$$
\begin{aligned}
\max \quad & f(\mathbf{x}) = CP_{\max}[\mathbf{x} = (chord(i), twist(i), i = 1, 17)]; \\
\text{s.t.} \quad & g_1(\mathbf{x}) = 9.5 - \lambda_{opt} > 0; \\
& g_2(\mathbf{x}) = CP_{\lambda=10} - CP_{0, \lambda=10} > 0; \\
& g_3(\mathbf{x}) = CP_{\lambda=13} - CP_{0, \lambda=13} > 0; \\
& g_j(\mathbf{x}) = chord(j-2) - chord(j-3) > 0 (j = 4, 5); \\
& g_j(\mathbf{x}) = chord(j-2) - chord(j-1) > 0 (j = 6, 18); \\
& g_j(\mathbf{x}) = twist(j-17) - twist(j-18) > 0 (j = 19); \\
& g_j(\mathbf{x}) = twist(j-18) - twist(j-17) > 0 (j = 20, 33); \\
& g_j(\mathbf{x}) = twist(j-17) - twist(j-18) > 0 (j = 34)
\end{aligned}
\tag{1}
$$

where, $\lambda$ is tip-speed ratio, $CP$ represents the power coefficient, $CP_{\max}$ represents the maximum power coefficient, $\lambda_{opt}$ represents the optimal tip-speed ratio corresponding to the maximum $CP$, the subscript "0" represents the performance parameters of the baseline blade. *Chord* and *twist* represent the chord length and twist angle of each control section of the blade, respectively. The numbers in brackets represent the serial number of the control section from root to tip, where "1" denotes the blade root, and "17" denotes the blade tip, as shown in Figure 13.

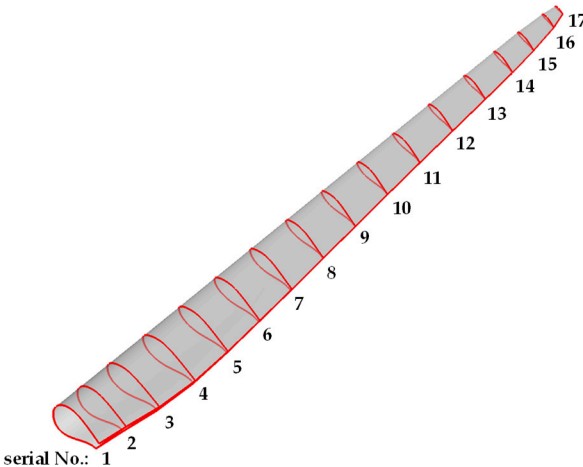

**Figure 13.** Serial number of the control section.

In Equation (1), the design goal is to maximize the maximum power coefficient ($CP_{max}$) so as to obtain the optimal wind energy capture capability, and the design variables are chord lengths and twist angles at 17 control sections; constraint $g_1(\mathbf{x})$ means that the optimal tip-speed ratio ($\lambda_{opt}$) should be less than 9.5; constraints $g_2(\mathbf{x})$ and $g_3(\mathbf{x})$ mean that the power coefficient of the design blade should be larger than that of the baseline blade at $\lambda = 10$ and $\lambda = 13$, which ensures a higher $CP$ at high tip-speed ratios; constraints $g_4(\mathbf{x})$ to $g_{18}(\mathbf{x})$ are used to constrain the chord length at each control section; constraints $g_{19}(\mathbf{x})$ to $g_{34}(\mathbf{x})$ are used to constrain the twist angle at each control section. It is noted that the reverse twist at the blade tip is helpful to reduce the induced drag and noise.

An in-house surrogate-based optimizer, "SurroOpt" [44] (Figure 14), was used for the optimization design of the wind turbine blade. The surrogate-based optimization refers to a method which uses surrogate models to replace time-consuming flow solvers, and refines the surrogate models by adding new sample points according to certain infill-sampling criteria, until the resulting "sequence of sample points" converges to the optimal solution. This method is proven to be efficient and robust, and more details about this method are presented in References [44–58]. The optimization process was as follows:

Step 1, "Start" in Figure 14. Determine the design space (variables and ranges). There were 34 design variables in the present study: 17 variables for the chord and 17 variables for the twist.

Step 2, "DoE" in Figure 14. The Latin Hypercube Sampling (LHS) method [52] was used to generate 40 or more initial sample points in the design space, which was called design of experiment (DoE). This method can avoid sample points clustering and make the sampling points more uniform. Each sample point corresponds to one distribution of the chord and one distribution of the twist.

Step 3, "Solver" in Figure 14. The BEM method was used to evaluate sample points to get the response functions of objectives and constraints, which are defined in "user interface" by the user according to Equation (1).

Step 4, "Construct surrogate models" in Figure 14. Kriging models were built for functions of objectives and constraints. Hyper parameters of the kriging models were tuned to improve the accuracy of prediction, corresponding to "Tune hyper parameters" in Figure 14.

Step 5, "Choose infill criteria" in Figure 14. Two infill criteria, i.e., expected improvement (EI) [57] and minimizing surrogate prediction (MSP) [58], were chosen to add new sample points. This process is called "sub optimization", which is the most crucial step of a surrogate-based optimization method.

Step 6, "Updating the sampled data" in Figure 14. The newly added sample points were evaluated by the BEM method to obtain the objective and constraints responses, and then added to the data set to update kriging models.

Step 7, "Main optimization" in Figure 14. Steps 4 to 6 were repeated until the termination conditions were reached. The termination conditions used in the present study were the maximum

number of sample points, accuracy of surrogate model, or distance between sample points. Once one of these termination conditions was reached, the optimization ended.

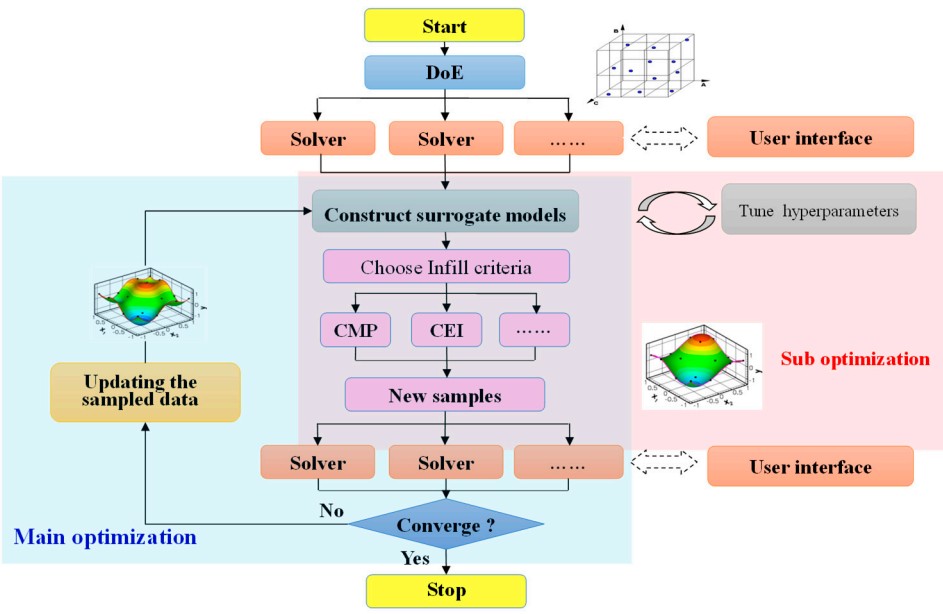

**Figure 14.** Framework of the efficient surrogate-based optimizer: SurroOpt.

It should be noted that in blade optimization, the main work of this paper is only to customize the objective function and constraint function related to the blade design in "user interface", but does not contribute to the framework or algorithm of SurroOpt.

## 4. Results and Discussion

### 4.1. Aerodynamic Design Optimization of Multi-Megawatt Wind Turbine Blade with Npu-Mwa Airfoils

The NREL 5 MW wind turbine blade was employed as the baseline, which is a 5 MW offshore wind turbine blade developed by Jonkman et al. in 2009 [17] (named NREL 5 MW blade), and the overall technical parameters are shown in Table 2.

**Table 2.** Overall technical parameters of the NREL 5 MW wind turbine [17].

| Parameter | Value |
|---|---|
| Rated power | 5 MW |
| Blade length | 61.5 m |
| Diameter of Wind mill | 126 m |
| Number of blades | three |
| Power adjustment mode | Variable-pitch and variable-speed |
| Cut-in wind speed | 3 m/s |
| Cut-out wind speed | 25 m/s |
| The minimum rotational speed of motor | 670 rpm |
| The rated speed of motor | 1173.7 rpm |
| The rated speed of wind mill | 12.1 rpm |
| Height of Hub | 90 m |
| Direction of rotation | Clockwise |
| Driving mode | Multi-stage gearbox |

For the NREL 5 MW blade, the airfoils with sharp trailing edge were used, including five DU airfoils (the relative thicknesses were 40%, 35%, 20%, 25%, and 21%) and one NACA airfoil (the relative

thickness was 18%), shown in Figure 15. The airfoil schedule of the NREL 5 MW blade is presented in Figure 16.

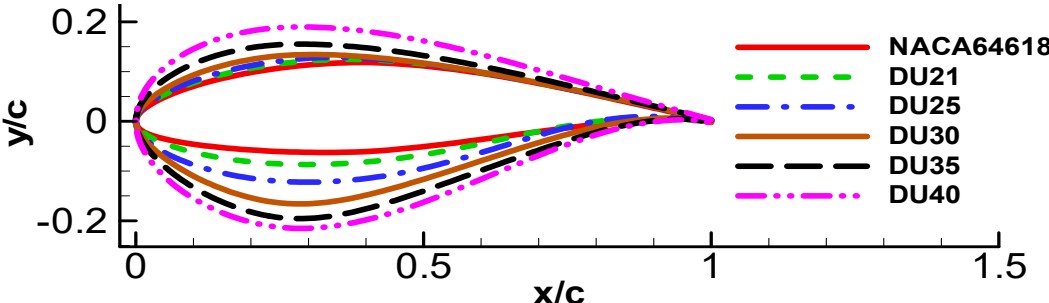

**Figure 15.** Geometric shape of airfoils adopted by the NREL 5 MW blade.

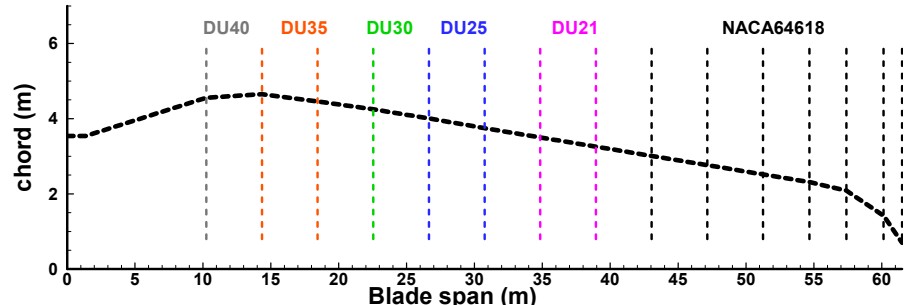

**Figure 16.** Span-wise airfoil arrangement of the NREL 5 MW wind turbine blade.

The aerodynamic design of a 5 MW wind turbine blade was carried out by using the newly developed NPU-MWA airfoils. The airfoils of each control section were firstly determined, and the span-wise airfoil arrangement is shown in Figure 17. The flat-back airfoils (NPU-MWA-600 and NPU-MWA-500) were adopted at the transition between the circular section and airfoil section, the NPU-MWA-400 and NPU-MWA-350 airfoils were used at the spanwise section of the maximum chord length, the NPU-MWA-250 and NPU-MWA-210 airfoils were selected as the main airfoil, and the NPU-MWA-180 airfoil was arranged at the blade tip.

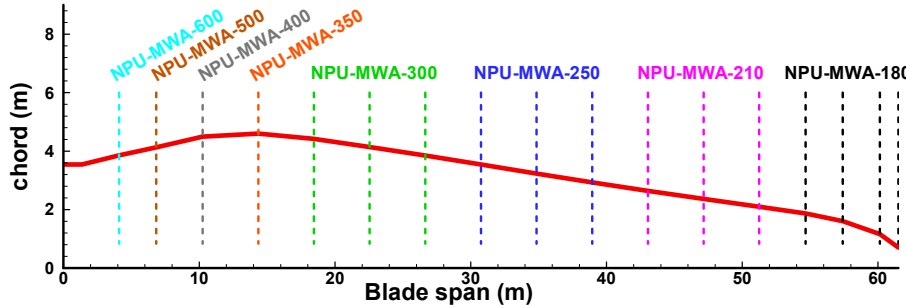

**Figure 17.** Span-wise airfoil arrangement of the designed NPU 5 MW wind turbine blade.

Figure 18 shows the convergence history of the optimization design. Figure 19 shows the optimized shape named NPU 5 MW blade. Figure 20 demonstrates the comparisons of geometric parameters between the NPU 5 MW blade and NREL 5 MW blade. It was noted that both the chord length and absolute thickness were reduced, which contributed to reducing the structural weight of the blade. This improvement was mainly due to the high design lift coefficient of NPU-MWA airfoils.

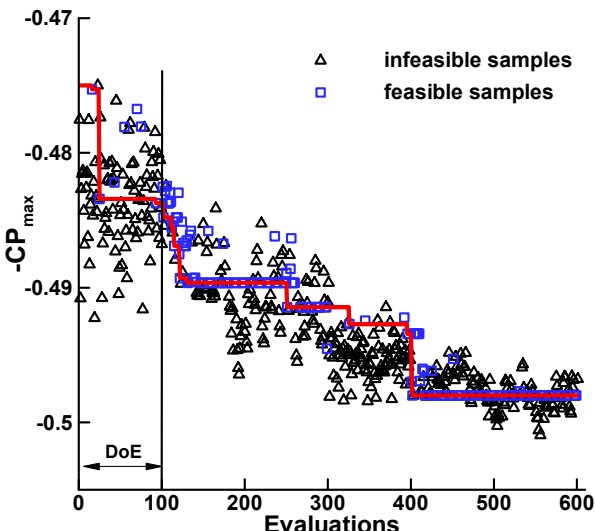

**Figure 18.** Convergence history of optimization design.

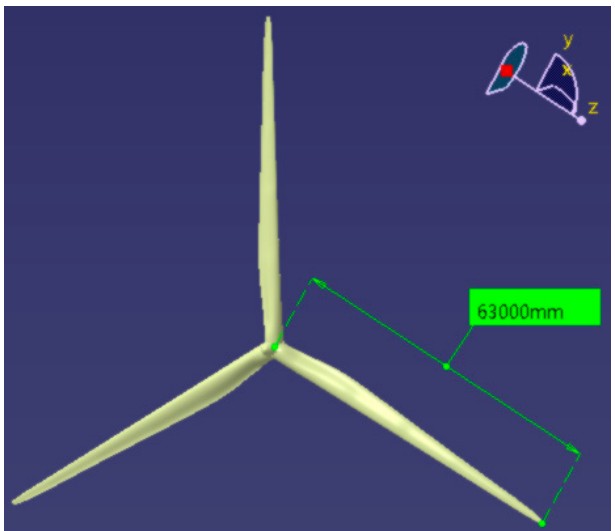

**Figure 19.** 3D model of the designed NPU 5 MW blade.

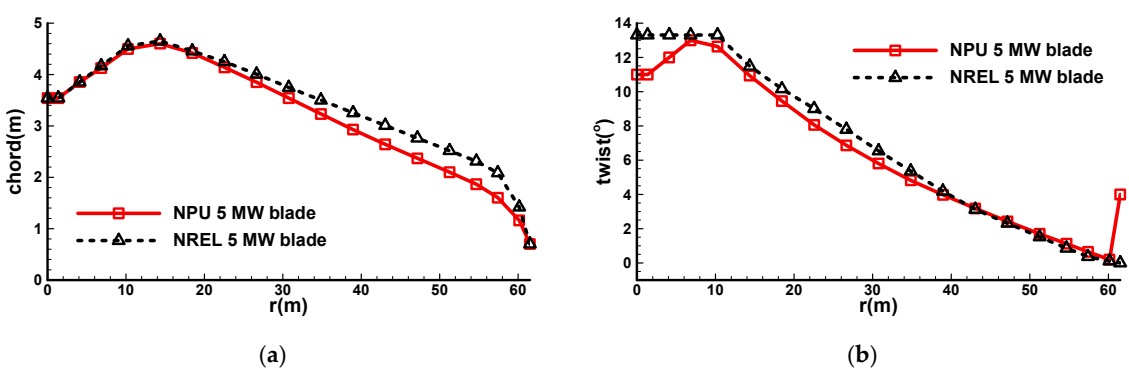

(**a**)                                                                             (**b**)

**Figure 20.** *Cont*.

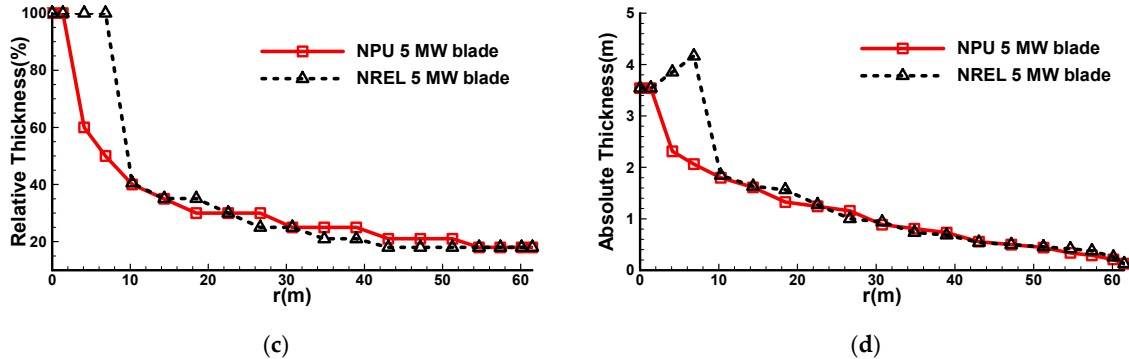

(**c**)                                     (**d**)

**Figure 20.** Comparison of geometric parameters between the NPU 5 MW blade and NREL 5 MW blade: (**a**) chord length distribution; (**b**) twist angle distribution; (**c**) relative thickness distribution; (**d**) absolute thickness distribution.

### 4.2. Aerodynamic Advantage of the NPU 5 MW Wind Turbine Blade with NPU-MWA Airfoils

The aerodynamic performances were compared between the NPU 5 MW blade and NREL 5 MW blade by the BEM, to demonstrate the aerodynamic advantage of NPU-MWA airfoils.

Figure 21 shows that the maximum power coefficient ($CP_{max}$) increased from 0.484 of the NREL 5 MW blade to 0.498 of the NPU 5 MW blade, and the corresponding optimal tip-speed ratio was 7.9. Also, $CP$ at high tip-speed ratios increased significantly. Figures 22 and 23 show that the rated wind speed of the NPU 5 MW blade was 11.25 m/s, which was lower than that of the NREL 5 MW blade. Moreover, higher power was produced by the NPU 5 MW blade at the same wind speed before reaching the rated condition. In addition, it is shown in Figure 24 that at the same wind speed, the thrust produced by the NPU 5 MW blade was smaller than that of the NREL 5 MW blade, which is beneficial to structural design and weight reduction.

The distributions of aerodynamic loads between the NPU 5 MW blade and NREL 5 MW blade are shown in Figure 25, with a rated wind speed of 11.25 m/s. Both the results by BEM and ROTNS can demonstrate the advantages of the NPU-MWA airfoil family, though the loadings by ROTNS were slightly higher than that of the BEM. According to the thrust distribution, the inboard thrust of the NPU 5 MW blade was larger than that of the NREL 5 MW blade because the NPU 5 MW blade uses flat-back airfoils with larger lift coefficients, and its chord length is almost the same as the NREL 5 MW blade (shown in Figure 20a). The outboard thrust of the NPU 5 MW blade was lower than that of the NREL 5 MW blade for two reasons: (1) more tangential force was generated due to the higher lift-to-drag ratio of the NPU-MWA airfoils; (2) the outboard chord length of the NPU 5 MW blade was significantly smaller than that of the NREL 5 MW blade. This reduction in thrust is helpful in reducing the displacement of the outboard blade, and reducing the stress level, thereby achieving a reduction in structural weight. Compared to the NREL 5 MW blade, the inboard thrust of the NPU 5 MW blade was increased, while the outboard thrust was reduced, and the total thrust of the NPU 5 MW blade was slightly lower. Due to the higher aerodynamic performance of the NPU-MWA airfoil family, the tangential force of the NPU 5 MW blade was higher than that of the NREL 5 MW blade, which is beneficial for generating more torque and improves the wind energy capture capability.

The abovementioned aerodynamic advantages of the NPU 5 MW blade are due to the higher design lift coefficient and higher lift-to-drag ratio of NPU-MWA airfoils.

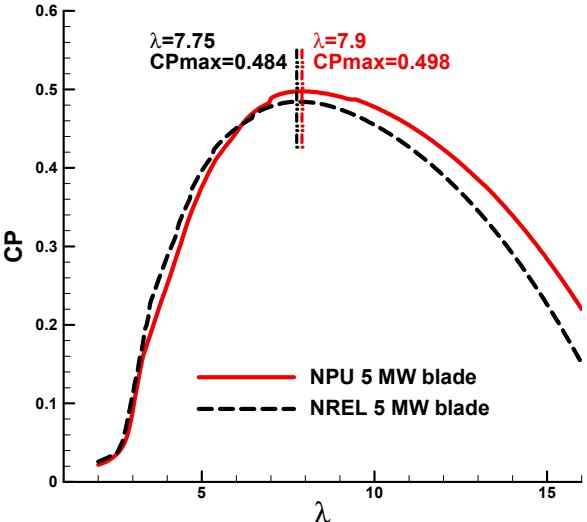

**Figure 21.** Comparison of power tip-speed (λ) ratio–coefficient (CP) for the NPU and NREL 5 MW blades.

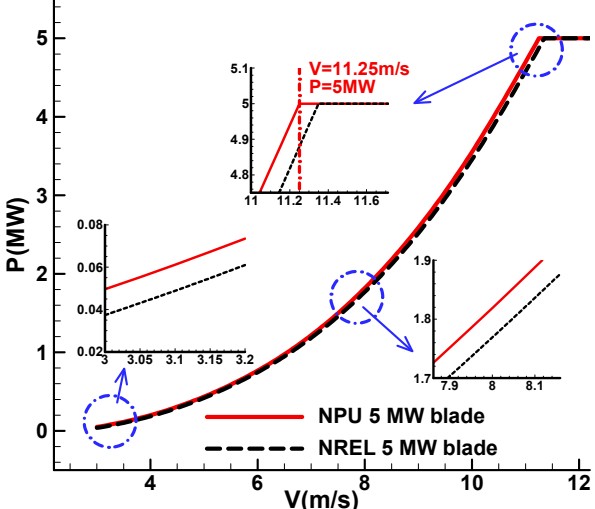

**Figure 22.** Comparison of wind speed (V)–power (P) for the NPU and NREL 5 MW blades.

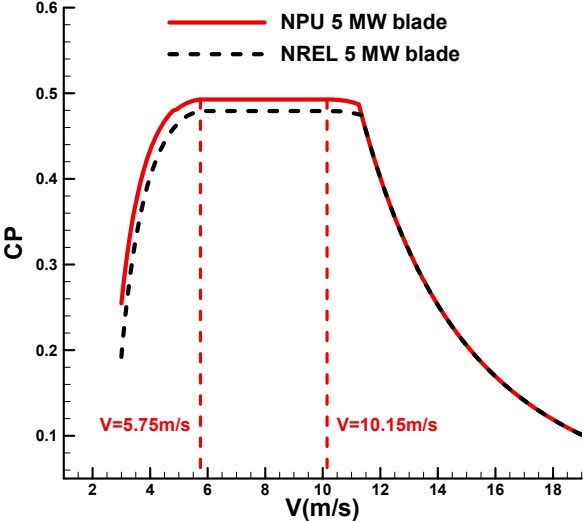

**Figure 23.** Comparison of power coefficient–wind speed for the NPU and NREL 5 MW blades.

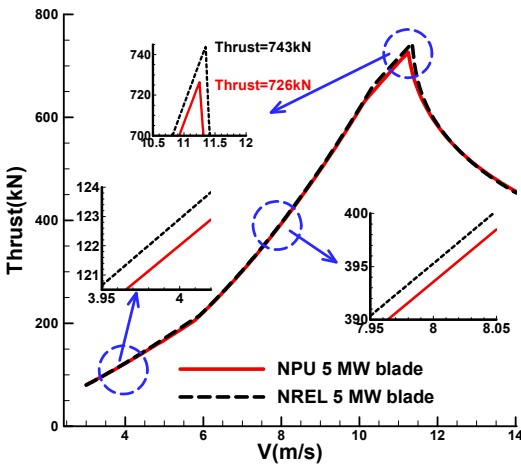

**Figure 24.** Comparison of thrust–wind speed for the NPU and NREL 5 MW blades.

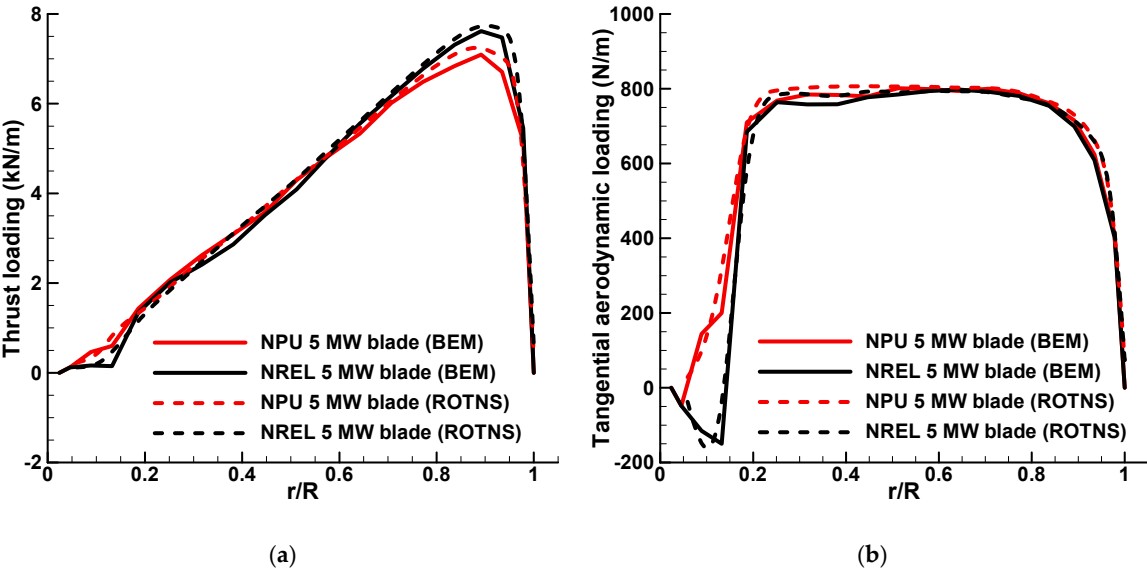

(**a**)                                        (**b**)

**Figure 25.** Distributions of aerodynamic loading: (**a**) thrust; (**b**) tangential force.

The in-house RANS solver, ROTNS, was adopted to examine the flow around the blade at different wind speeds, ranging from 3 m/s to 11 m/s. The results show that the NPU 5 MW blade can produce more power than that of the NREL 5 MW blade at the same wind speed, as shown in Figure 26.

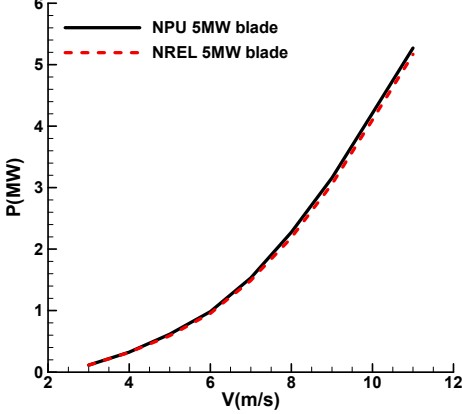

**Figure 26.** Comparison of power generation between the NREL 5 MW and NPU 5 MW by ROTNS.

Figure 27 presents the surface streamlines of the NPU 5 MW blade at different wind speeds. The pressure coefficient Cp is defined as $\frac{p-p_\infty}{0.5\rho_\infty\left(V_\infty^2+\Omega^2r^2\right)}$, where $p_\infty$ and $\rho_\infty$ are the ambient pressure and density, $V_\infty$ the wind speed, $\Omega$ the angular velocity of the blade, and r the local radius. The local suction peaks are shown obviously. It was demonstrated that no separation occurred in the suction side of the blade, which is related to the operating mode of the wind turbine. At a low wind speed (>3 m/s), the minimum rotational speed of the blade was limited by the generator, and remained unchanged at a wind speed lower than 5.75 m/s. Thereafter, the rotational speed of the blade increased as the wind speed increased to the rated wind speed of 11.25 m/s, so as to keep the optimal tip-speed ratio unchanged. Furthermore, at a wind speed of 3 m/s, the effective angles of attack for most sections ranged from −2°–0°. At these small angles of attack, no large separation occurs. At a wind speed of 7 m/s to 11 m/s, the effective angle of attack for most sections ranged from 4°~8°, corresponding to the maximum lift-to-drag ratio, and thus there was also no large separation flow. This is not an improvement of the NPU blade. In fact, the aerodynamic performance of the NREL blade is also very good. It should be noted that although the S-A turbulence model is not suitable for large separation flows, the attached flow or small separation flow in the example can be reasonably simulated by the S-A model.

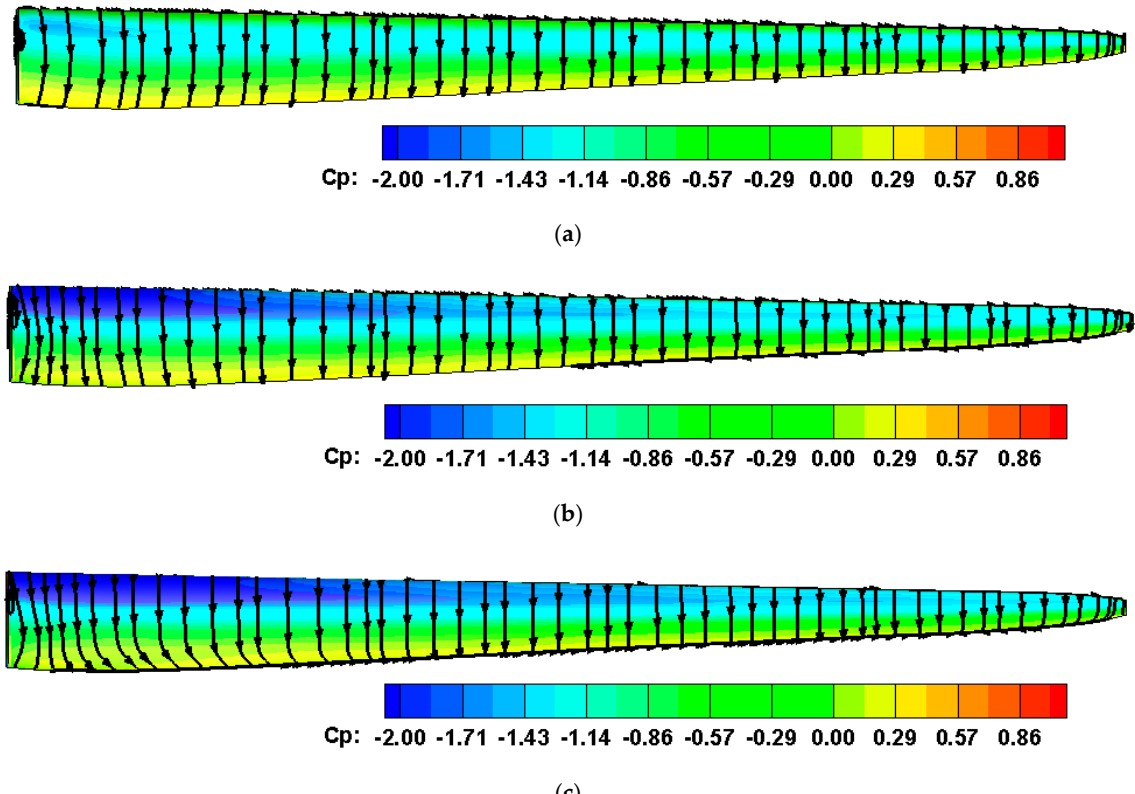

**Figure 27.** Surface streamlines of the NPU 5 MW blade at different wind speeds: (**a**) wind speed of 3 m/s; (**b**) wind speed of 7 m/s; (**c**) wind speed of 11 m/s.

In summary, compared to the NREL 5 MW blade, the NPU 5 MW blade with NPU-MWA airfoils produced more power but smaller thrust at the same wind speed over a wide range of operating conditions. This improvement in aerodynamic performance is mainly due to higher lift-to-drag ratio of NPU-MWA airfoils.

*4.3. Structural Advantage of the NPU 5 MW Wind Turbine Blade with NPU-MWA Airfoils*

In this section, the structural performance of the NPU 5 MW wind turbine blade is analyzed, using the same material as the NREL 5 MW blade, presented in Reference [18]. Table 3, created referring to

Reference [18], shows the properties of materials. D means uni-directional material and DB denotes double-bias material. Table 4 shows the mapping of stacks and materials. The stack ID from 1 to 7 are for skin layup on the blade surface, and the stack ID from 8 to 9 are for shear web layup.

**Table 3.** Properties of material used in the 5 MW wind turbine blade (created referring to [18]).

| | Layer Thickness (mm) | Ex (MPa) | Ey (MPa) | Gxy (MPa) | Prxy (-) | Dens. (kg/m³) | UTS (MPa) | UGS (MPa) |
|---|---|---|---|---|---|---|---|---|
| Gelcoat | 0.05 | 3440 | / | 1380 | 0.3 | 1235 | / | / |
| E-LT 5500 (UD) | 0.47 | 41,800 | 14,000 | 2630 | 0.28 | 1920 | 972 | 702 |
| Triax | 0.94 | 27,700 | 13,650 | 7200 | 0.39 | 1850 | 700 | / |
| Saertex (DB) | 1 | 13,600 | 13,300 | 11,800 | 0.49 | 1780 | 144 | 213 |
| FOAM | 1 | 256 | 256 | 22 | 0.3 | 200 | / | / |
| Carbon (UD) | 0.47 | 114,500 | 8390 | 5990 | 0.27 | 1220 | 1546 | 1047 |

**Table 4.** Mapping of stacks and materials (created referring to [18]).

| Stack ID | Stack Name | Material |
|---|---|---|
| 1 | Gelcoat | Gelcoat |
| 2 | Triax-Skins | Triax |
| 3 | Triax-Root | Triax |
| 4 | UD-Carbon | Carbon (UD) |
| 5 | UD-Glass-TE | E-LT-5500 (UD) |
| 6 | TE-Foam | FOAM |
| 7 | LE-Foam | FOAM |
| 8 | / | Saertex (DB) |
| 9 | / | FOAM |

Figure 28 demonstrates the finite element mesh for structural analysis, which had 11,597 grid cells and 11,080 grid points. Also, the panel models of the cross section are shown, where A is the leading edge (LE), B is the sandwich panel of the leading edge (LE-SP), C is the spar cap, D is the sandwich panel of the trailing edge (TE-SP), E is the reinforcement panel of the trailing edge (TE-RP), F is the trailing edge, and G is the shear webs.

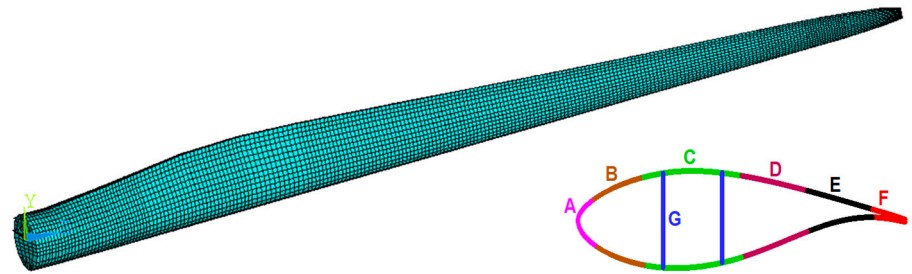

**Figure 28.** Finite element mesh and panel models of cross section (A: leading edge (LE), B: sandwich panel of leading edge (LE-SP), C: spar cap, D: sandwich panel of trailing edge (TE-SP), E: reinforcement panel of trailing edge (TE-RP), F: trailing edge (TE), G: shear webs).

Referring to Reference [18], the stack ID in each panel and shear webs are shown in Tables 5 and 6. The layup designs of the NREL 5 MW blade and the NPU 5 MW blade are shown in Figures 29 and 30, respectively. The difference in the layup design of the NPU 5 MW blade over that of the NREL 5 MW blade was mainly at the trailing edge of the blade root, in which the NPU 5 MW blade with flat-back airfoil had a thinner layer than that of the NREL 5 MW blade with sharp-trailing edge airfoil. The reduced materials were Triax and E-LT-5500 (UD). The goal of the layup design was to maintain considerable stiffness and strength with the NREL blade. It should be noted that no optimization was



carried out, the layup design was changed by "cut and try" until the blade tip displacement and stress level of the NPU 5 MW blade were comparable to those of the NREL 5 MW blade.

**Table 5.** Stack usage (Stack ID) in each panel along blade span (created referring to Reference [18]).

| Blade Span (m) | A: LE | B: LE-SP | C: Spar Cap | D: TE-SP | E: TE-RP | F: TE |
|---|---|---|---|---|---|---|
| 0 | 1, 2, 3, 2 | 1, 2, 3, 7, 2 | 1, 2, 3, 4, 2 | 1, 2, 3, 6, 2 | 1, 2, 3, 5, 6, 2 | 1, 2, 3, 2 |
| 1.37 | 1, 2, 3, 2 | 1, 2, 3, 7, 2 | 1, 2, 3, 4, 2 | 1, 2, 3, 6, 2 | 1, 2, 3, 5, 6, 2 | 1, 2, 3, 2 |
| 6.83 | 1, 2, 3, 2 | 1, 2, 3, 7, 2 | 1, 2, 3, 4, 2 | 1, 2, 3, 6, 2 | 1, 2, 3, 5, 6, 2 | 1, 2, 3, 2 |
| 10.25 | 1, 2, 2 | 1, 2, 7, 2 | 1, 2, 4, 2 | 1, 2, 6, 2 | 1, 2, 5, 6, 2 | 1, 2, 2 |
| 43.05 | 1, 2, 2 | 1, 2, 7, 2 | 1, 2, 4, 2 | 1, 2, 6, 2 | 1, 2, 5, 6, 2 | 1, 2, 2 |
| 47.15 | 1, 2, 2 | 1, 2, 7, 2 | 1, 2, 4, 2 | 1, 2, 6, 2 | 1, 2, 6, 2 | 1, 2, 2 |
| 57.40 | 1, 2, 2 | 1, 2, 7, 2 | 1, 2, 4, 2 | 1, 2, 6, 2 | 1, 2, 6, 2 | 1, 2, 2 |
| 60.00 | 1, 2, 2 | 1, 2, 2 | 1, 2, 2 | 1, 2, 2 | 1, 2, 2 | 1, 2, 2 |
| 61.50 | 1, 2, 2 | 1, 2, 2 | 1, 2, 2 | 1, 2, 2 | 1, 2, 2 | 1, 2, 2 |

**Table 6.** Stack usage (Stack ID) in shear webs (created referring to Reference [18]).

| Blade Span (m) | Shear Webs | Number of Layers of DB per Stack | FOAM Thickness |
|---|---|---|---|
| 1.37 | 8,9,8 | Two | 50 mm |
| 61.50 | 8,9,8 | Two | 50 mm |

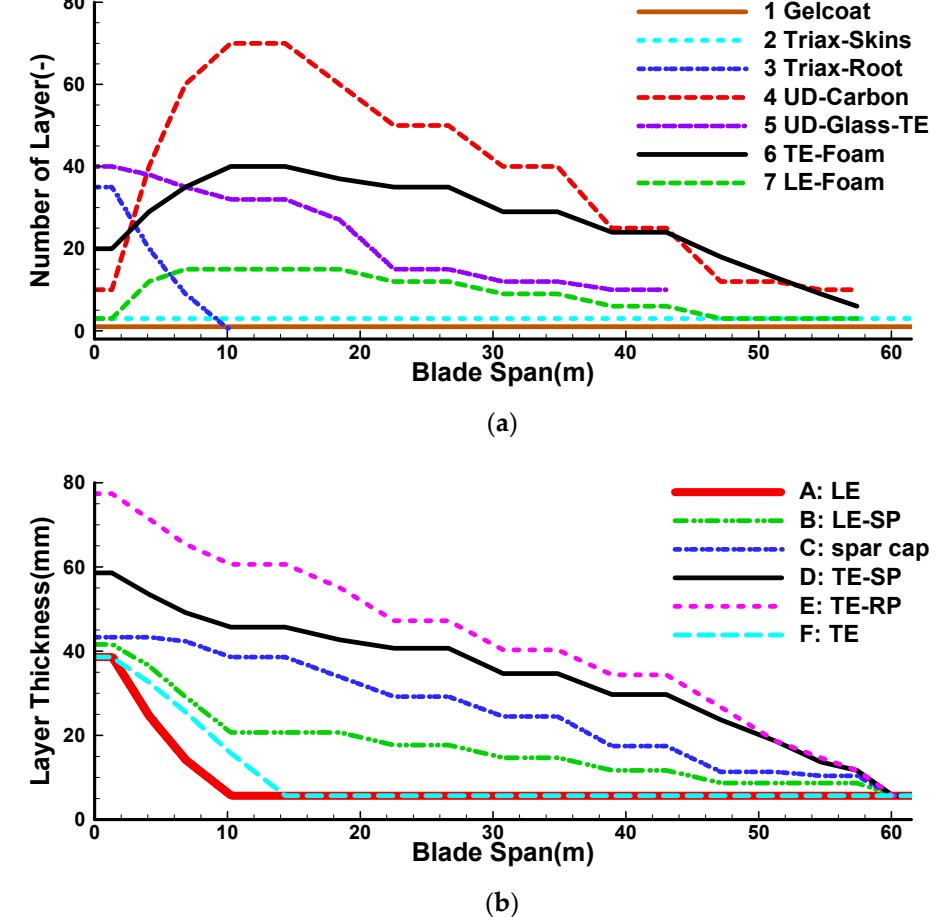

**Figure 29.** Layup design of the NREL 5 MW blade: (**a**) number of layers; (**b**) thickness of layer.

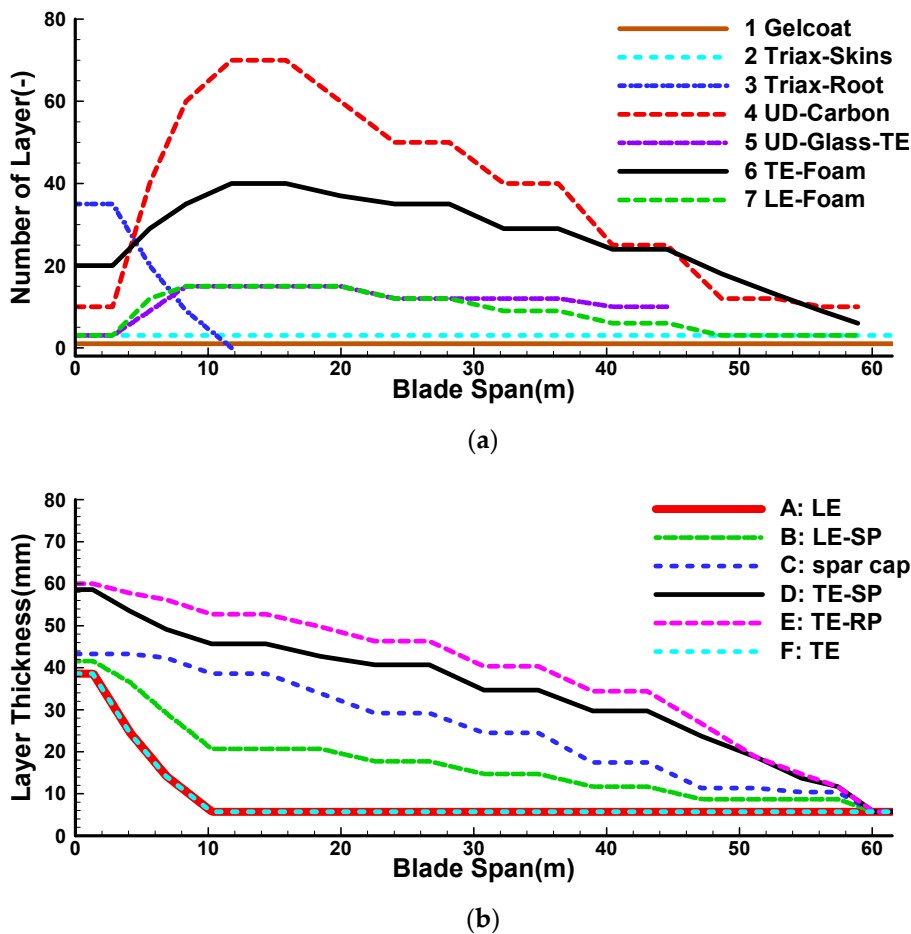

(**a**)

(**b**)

**Figure 30.** Layup design of the NPU 5 MW blade: (**a**) number of layers; (**b**) thickness of layer.

ROTNS was adopted to obtain the aerodynamic loads for structural analysis at four typical working conditions, including cut-in (V = 3 m/s), optimal tip-speed ratio (V = 7 m/s), rated power (V = 11 m/s), and cut-out (V = 25 m/s). The finite element model was used to analyze the structural performance of wind turbine blades. Figures 31 and 32 show that the stiffness (maximum displacement) and strength (maximum stress) characteristics of the NPU 5 MW blade were almost the same as those of the NREL 5 MW blade.

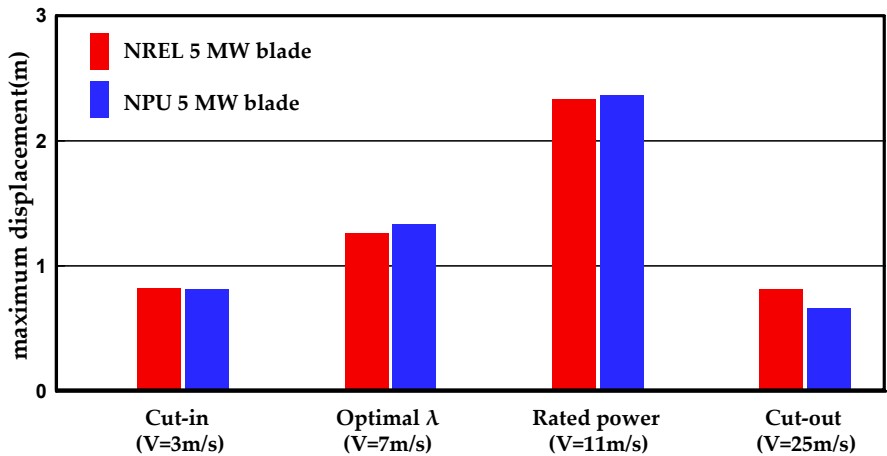

**Figure 31.** The maximum displacement of blade tip at different working conditions.

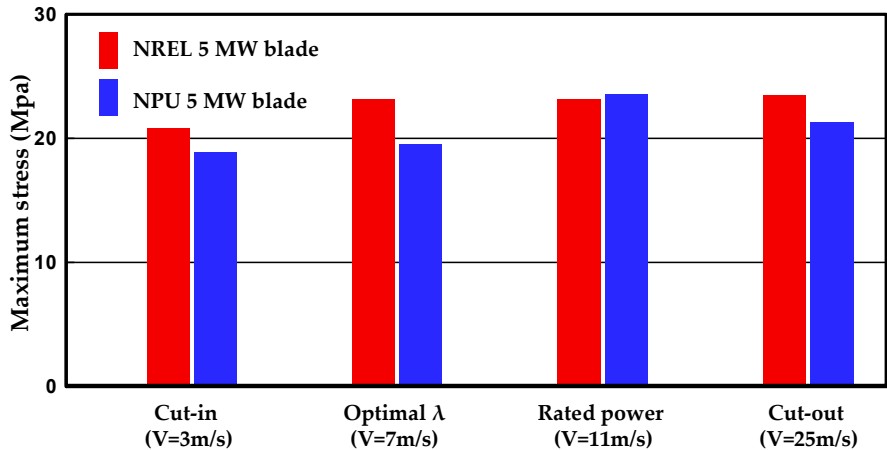

**Figure 32.** The maximum stress of blade tip at different working conditions.

Moreover, the natural frequencies and vibration modes of the NPU 5 MW blade were examined at 0 rpm, and shown in Table 7. At the rated rotational speed, obviously the natural frequencies of the blade avoid the resonance point perfectly. The frequency of the first-order flapwise was 22% higher than the triple frequency of rotation, which satisfies the suggestion by DNV (DET NORSKE VERITAS) /Riso [59] that the natural frequency should be outside the range of ±12% of the frequency multiplication. According to the Campbell diagram of the NREL 5 MW blade given in Reference [60], the natural frequencies of the blade varied slightly with the rotational speed, and the natural frequency at the rated speed was slightly higher than that under static conditions. Therefore, the natural frequencies of the NPU blade at the rated speed would be higher, which is better for avoiding the resonance point.

**Table 7.** Rotational frequencies at rated speed (12.1 rpm) and natural frequencies of the NPU 5 MW blade at 0 rpm.

|  | 1P | 2P | 3P | 6P | 9P | 1st Flapwise | 2nd Edgewise | 2nd Flapwise |
|---|---|---|---|---|---|---|---|---|
| Frequency (Hz) | 0.202 | 0.404 | 0.607 | 1.213 | 1.820 | 0.74 | 0.91 | 1.54 |

Table 8 shows that compared with the NREL 5 MW blade, the NPU 5 MW blade had a weight reduction of 9% and the spanwise location of centroid was closer to the blade root, which helps to increase the natural frequency.

**Table 8.** Comparison of mass characteristics among three 5 MW blades.

|  | Weight of Blade (kg) | Spanwise Location of Centriod (m) |
|---|---|---|
| NREL 5 MW blade | 16639 | 24.373 |
| NPU 5 MW blade | 15153 | 23.479 |
| Δ | −8.93% | −3.67% |

The structural analysis results show that the stiffness and strength of the designed NPU 5 MW blade were equivalent to the NREL 5 MW blade under all of the working conditions. The results of frequency analysis show that the NPU 5 MW blade can be used safely. The structural weight of the NPU 5 MW blade was significantly lower than that of the NREL 5 MW blade, which is an exciting result. There are two reasons for the reduction in the weight of the NREL 5 MW blade: the first one is that the chord of blade section is reduced under the same aerodynamic loads due to high design lift coefficients of NPU-MWA airfoils; the second one is that both the number and thickness of layers at the blade root are reduced by using flat-back airfoils. The flat-back airfoils had better performance in

structural than sharp-trailing edge airfoils under the same layer design, fewer materials are used to maintain the same structural performance, and thus the structural weight of the blade is reduced.

Although only the natural frequencies and vibration modes were examined in the present study, the current results are sufficient to support the advantages of the NPU-MWA airfoil family in the structural reduction of multi-megawatt wind turbine blades. In the future, analyses will be conducted considering more factors, such as a real wind turbine, where all three blades can feel the presence of the other blades through a soft shaft, the whirling of the wind turbine, and also the frequencies split up in a forward and a backward whirling [61,62].

## 5. Conclusions

A 5 MW wind turbine blade was optimized by using the newly developed NPU-MWA airfoils. Aerodynamic performance and structural characteristics were analyzed to demonstrate the advantages of NPU-MWA airfoils. The following conclusions can be drawn:

(1) Due to the higher lift-to-drag ratio of NPU-MWA airfoils, the maximum power coefficient increased from 0.484 of the NREL 5 MW blade to 0.498 of the NPU 5 MW blade, and the rated wind speed of the NPU 5 MW blade was lower than that of the NREL 5 MW blade. The NPU 5 MW blade produced more power but a smaller thrust at the same wind speed over a wide range of operating conditions. A smaller thrust is beneficial to structural design and weight reduction. The improvements in the aerodynamic performance of the NPU 5 MW blade were validated by the RANS solver;

(2) With the same level of structural stiffness and strength, the NPU 5 MW blade was 9% lighter than the NREL 5 MW blade. One reason is that high design lift coefficients of NPU-MWA airfoils help to reduce chord of blade under the same aerodynamic loads, resulting in a lighter structural weight for a constant layer design; the other reason is that flat-back airfoils are adopted inboard of the blade to improve the structural performance, resulting in a thinner layer design for the same stiffness and strength.

**Author Contributions:** Conceptualization, Z.H. and W.S.; Data curation, X.Y.; Formal analysis, J.X. and X.Y.; Funding acquisition, J.X. and W.S.; Investigation, J.X. and X.Y.; Methodology, J.X. and Z.H.; Project administration, W.S.; Resources, J.X. and X.Y.; Software, J.X. and Z.H.; Supervision, W.S.; Validation, J.X.; Visualization, J.X. and X.Y.; Writing—Original draft, J.X.; Writing—Review & editing, J.X., Z.H. and W.S.

**Funding:** This research was funded by the National Natural Science Foundation of China (NSFC) (Grant number 11772261 and 11972305), the Funds for National Key Laboratory of Science and Technology on Aerodynamic Design and Research (Grant number JCKYS2019607005 and 2018KC010116), the National High Technology Research and Development Program ("863" Program) of China (Grant number 2012AA051301) and the Fundamental Research Funds for the Central Universities of Northwestern Polytechnical University (Grant number 310201401JCQ01017).

**Acknowledgments:** The authors want to thankfully acknowledge Han Nie and Mingwei Ge for their contributions of improving the English expression of this article.

**Conflicts of Interest:** The authors declare no conflict of interest.

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
