# Peer review of "Design Optimization of a Multi-Megawatt Wind Turbine Blade with the NPU-MWA Airfoil Family"

_energies, doi:10.3390/en12173330_

Round 1

Reviewer 1 Report

The aim of the paper is to show that the NREL 5MW can be improved by using a new airfoil family denoted NPU-MWA by reducing the weight approximately 9%.

Page 2 line 54  is written that wind turbine blades using dedicated airfoil families have lower loads. Is that really true since a certain load is needed to capture efficiently the power in the wind and the aim of the blades are to provide this load. Please explain more what you mean by reduced loads.

It is considered very positive that the aerodynamic coefficients were verified by actual wind tunnel measurements. A nice description of the design philosophy of the NPU-MWA family is written, but this design was made in 2016 and therefore not the core of this paper. A note could be written that the stall of the 21% thick airfoil is much more abrupt that the DU airfoil (Figure 6). But again, the airfoil design was already made before this paper.

The blade aerodynamics is based on BEM and a FV CFD code using a Spalart Allmaras turbulence model. The CFD was validated against the NREL Phase VI experiment. In Figure 11 the reference should be 35 and not 33 !!!!!!!

Eq. 1 and Figure 15 showing the optimizer algorithm is very difficult to understand and a better description must be given. This continues on page 10, so please describe your algorithm more clearly.

Figure 20 shows the convergence of the optimizer and Figure 23 shows the final Cp(lambda) curve. A 3% increase in Cp,max is reported together with a generally higher Cp for high tip speed ratios, which comes from the better performance (higher cl/cd) of the NPU-MWA airfoils. The decrease in thrust shown in Figure 26 is marginal, and could be within the uncertainty margin. A certain thrust force is needed to produce a certain power, but it could be that the slightly lower thrust comes from the lower chord at the innermost radius not contributing very much to the torque. To explain this one should plot the normal load distributions e.g. at Vrated and compare with the one from the NREL 5MW rotor.

Figure 28 shows computed streamlines on the suction side of the NPU 5MW blade for 3, 7 and 11 m/s and it is written that no or very little flow separation occurs. Is this also true if the NREL rotor was computed using the same code and turbulence model, or is it an improvement of the new rotor. Please also explain what is meant by P/P_infinity – the range between 0.99 and 1.0025 is extremely small and is certainly not the ratio between the surface pressure and the ambient pressure.

What determines the structural layout of the NPU blade ? The layout is shown and compared to the original NREL blade, but the paper does not mention what are the load cases used to determine the layup and was the final layup determined using the optimizing algorithm. Please explain. The result is a lighter blade, mostly coming from lowering the chord due to using flatback airfoils near the root and higher design lift for the outer part. Taking all uncertainties into account is it fair to give the reduction with two decimals 8.93 %, perhaps one should instead state 8.9 or 9 %. The same goes for the natural frequencies that are stated using 4 decimals.

In Figure 34 is shown a Campbell diagram. This is for one blade only, but please note that on a real wind turbine where all three blades can feel the presence of the other blades through a soft shaft the turbine will experience whirling and the frequencies will split up in a forward and a backward whirling, e.g. the 1 flapwise will split up in omega_flap+omega_rotor and omega_flap-omega_rotor and not constants lines as shown in the figure.

Author Response

Dear Editor and Reviewers,

We would like to thank you for reviewing our paper and giving us thorough and thoughtful comments. According to your valuable suggestion, we have made great efforts to revise the manuscript. Also, we have addressed the reviewers’ comments one by one, and the revised text is highlighted in red color in the new version of manuscript.

Many thanks!

Reviewer 2 Report

The paper perform the aerodynamic design of a horizontal axis wind turbine based on a new family of airfoils. The design is based on simulation based optimization and shows an improvement both in performance and in associated weight. The paper is relevant to the field and is well structured. I recommend the publication of this manuscript once the few minor items below are addressed.

What is the maximum Reynolds number that the blades of new designed turbine experiences? Is that Reynolds number within the range of the validation performed in section 3.1 and experimentally in section 2?

Is the Cp calculations in Figure 23 performed with the same software for both turbines?

The discussion of the separation associated with Figure 28 may not be accurate as the SA turbulence model cannot capture separation accurately. The authors need to add comments to explain this weakness of the model.

Author Response

(The authors gave the same response as above.)

Round 2

Reviewer 1 Report

The main issue of the paper is to show that the NPU-MWA airfoil family is more efficient than the DU family in the sense that a lighter blade can be achieved and increase slightly Cp,max. The more efficient airfoils(higher l/d) are capable of producing the necessary loads at smaller chords and that is the main reason for lowering the weight. Also the normal force distribution shows a decrease at the outer part, which is beneficial for lowering the root bending moments. CFD is used to investigate the flow past the optimized blade using the new airfoils.

It would be nice to also include the load distribution from the CFD in Figure 25 to compare with the BEM.

The ratio p/p_inf close to 1 just verifies that the pressure gradient normal to the surface in a boundary layer is small. I suggest instead to show cp=(po-p)/0.5*density*(vo^2+omega^2*r^2), where po is the ambient pressure and omega the angular velocity of the blade and r the local radius. This will show the local suction peaks.

The structural part of the paper is not very important, except that the more slim blade reduces the mass and thus also change the natural frequencies.

Author Response

Dear Editor and Reviewers,

We would like to thank you for reviewing our paper and giving us thorough and thoughtful comments. According to your valuable suggestion, we have made great efforts to revise the manuscript. Also, we have addressed the reviewers’ comments one by one, and the revised text is highlighted in blue color in the new version of manuscript.

Many thanks!
